# Synthesis of New AIEE-Active Chalcones for Imaging of Mitochondria in Living Cells and Zebrafish In Vivo

**DOI:** 10.3390/ijms22168949

**Published:** 2021-08-19

**Authors:** Huiqing Luo, Na Li, Liyan Liu, Huaqiao Wang, Feng He

**Affiliations:** 1School of Pharmaceutical Science, Sun Yat-sen University, Guangzhou 510006, China; luohq8@mail2.sysu.edu.cn (H.L.); lina49@mail2.sysu.edu.cn (N.L.); liuly37@mail2.sysu.edu.cn (L.L.); 2Department of Anatomy and Neurobiology, Zhongshan School of Medicine, Sun Yat-sen University, Guangzhou 510006, China; wanghq@mail.sysu.edu.cn

**Keywords:** AIEE, chalcone, mitochondria, zebrafish

## Abstract

Fluorophores with aggregation-induced emission enhancement (AIEE) properties have attracted increasing interest in recent years. On the basis of our previous research, we successfully designed and synthesized eleven chalcones. Through an optical performance experiment, we confirmed that compounds **1**–**6** had obvious AIEE properties. As these AIEE molecules had excellent fluorescence properties and a large Stokes shift, we studied their application in living cell imaging, and the results showed that these compounds had low cytotoxicity and good biocompatibility at the experimental concentrations. More importantly, they could specifically label mitochondria. Subsequently, we selected zebrafish as experimental animals to explore the possibilities of these compounds in animal imaging. The fluorescence imaging of zebrafish showed that these AIEE molecules can enter the embryo and can be targeted to aggregate in the digestive tract, which provides a strong foundation for their practical application in the field of biological imaging. Compared with traditional fluorophores, these AIEE molecules have the advantages of possessing a small molecular weight and high flexibility. Therefore, they have excellent application prospects in the field of biological imaging. In addition, the findings of this study have very positive practical significance for the discovery of more AIEE molecules.

## 1. Introduction

Organic fluorescent dyes have been generally applied to biological imaging, fluorescent probes, pathological detection, and other fields on account of their strong fluorescence emission [1,2,3,4]. However, traditional organic fluorescent dyes often show an aggregation-caused quenching (ACQ) phenomenon, which severely limits their practical application. ACQ molecules emit strong fluorescence in low-concentration solutions, but show fluorescence quench in high concentrations or solid state [5]. Tang’s research group first discovered a compound with an aggregation-induced emission (AIE) phenomenon called 1-methyl-1,2,3,4,5-pentaphenylsilole in 2001 [6]. Fluorescence molecules with this property show weak emission in solution but show a dramatically enhanced emission in high concentrations and solid state, which is contrary to the ACQ phenomenon [7]. The AIEE phenomenon has attracted many scientists interest since it was first discovered. An increasing number of AIEE molecules have been found with the development of research, including derivatives from tetraphenylethylene (TPE), 2-phenylcinnamylnitrile, and hexaphenylsilole (HPS) [8,9,10]. The causes of the AIEE phenomenon have been extensively studied, and various mechanisms have been proposed, including restricted intramolecular rotation (RIR), J-type aggregation, restriction of intramolecular vibration (RIV), twisted intramolecular charge transfer (TICT), and Z/E isomerization. Of all the above factors, RIR is considered to be the main cause [11,12,13,14,15,16]. Since the AIEE phenomenon was proposed, many compounds with this property have been synthesized and successfully applied in the biomedical field in the past decades, mainly including in the monitoring of life activities, the diagnosis of diseases, and the treatment of diseases [17,18,19,20].

In our previous studies, as shown in Figure 1, we researched the fluorescence properties of sanguinarine and found that it showed obvious AIEE properties [21]. Molecules with a large conjugate system and high molecular weights are easily inserted into DNA bases and cause toxicity. Simultaneously, according to Lipinski’s rules of five, they will be greatly limited in practical applications in medical research [22,23]. Therefore, we tried to reduce the planarity and the conjugated structure by breaking the oxygen-containing pentacyclic ring at the end of the sanguinarine. The experiments results showed that chelerythrine was also a typical AIEE molecule [21]. Subsequently, on the basis of chelerythrine, we synthesized six flavonoid and two flavanone compounds by opening another benzene ring. They also exhibited AIEE characteristics [24,25]. In addition, chalcone-based AIEE molecules have been reported in several works [26,27,28], which further increases our interest and confidence in the study of the fluorescence characteristics of chalcone compounds. In this project, we tried to acquire smaller molecules through the same ideas and experiences. Fortunately, we eventually obtained 11 chalcones (chemical structure is shown in Figure 1) and confirmed that compounds **1**–**6** had AIEE properties by measuring their optical properties. In Table 1, we calculated the molecular weight (MW), partition coefficient (logp), and number of hydrogen bond acceptors (nON) of all the chalcone derivatives.

Chalcones are a kind of natural product that widely exist in nature and possess many pharmacological activities, including anti-infection, anti-inflammatory, anti-cancer, antioxidant, anti-bacterial and anti-viral properties [29,30,31,32,33,34,35,36]. In this paper, 11 target products were obtained through the Claisen–Schmidt reaction. By studying their optical properties, such as the fluorescence spectrum, UV spectrum, fluorescence images under 365 nm UV light, fluorescence quantum yield (ϕF), and fluorescence stability, compounds **1**–**6** were concluded to be AIEE molecules. Subsequently, through different viscosity tests, it was speculated that the cause of the AIEE phenomenon was restricted intramolecular rotation (RIR). Due to the fact that they present strong emission and stable fluorescence intensity characteristics, we tried to explore the possibilities in terms of biological imaging. Using an MTT assay, a cell uptake assay, and a mitochondrial co-localization assay, chalcones were identified as having good biocompatibility and labeling the mitochondria specifically under experimental concentrations. After zebrafish embryos were infiltrated by compounds **1**–**6** for 1 h, bright blue fluorescences were observed at 24, 48, and 120 h, respectively. Additionally, we found that they had different affinities for the tissues and organs of zebrafish and specifically gathered in the digestive tract, which indicates that AIEE compounds **1**–**6** have great potential for practical application in zebrafish imaging.

## 2. Results and Discussion

### 2.1. Optical Properties and Aggregation-Induced Emission Enhancement Properties

Compounds **1**–**11** are all soluble in methanol but insoluble in water. In order to investigate their fluorescence properties, we configured CH_3_OH/H_2_O mixed solutions with different water contents (0–90%), respectively, and measured the emission values at room temperature by PL spectra. As shown in Figure 2a, the PL intensity of compound **1** was relatively low in a pure CH_3_OH solution. As the water contents increased in the mixed solution, the PL intensity gradually increased and reached the maximum when the volume fraction of water was 70%. As can been seen in Appendix A, the maximum fluorescence intensity of compound 1 with 70% water content was about 15-fold of that in the pure CH_3_OH solution, indicating that compound **1** exhibits typical AIEE properties [13]. However, as the water fractions increased, the fluorescence intensity showed a tendency to decrease. A possible explanation for this phenomenon is that when water, a poor solvent, was added, the solute molecules could aggregate into two forms: regular nanoparticles and amorphous particles. The regular nanoparticles led to enhanced fluorescence intensity, while the amorphous particles resulted in a decrease in fluorescence intensity [37]. In addition, as shown in Appendix A, there was a large Stokes shift, which is favorable for its fluorescent labeling properties [38], of about 164 nm. A similar variation trend can be observed in Figure 2b; the maximum PL intensity was about 11-fold of that in 0%. As shown in Figure 2c, the PL intensity gradually increased when the volume fraction of water changed from 0% to 80%, and it reached a maximum when the water content was 80%, with an approximate 11-fold increase compared with the pure methanol solution. As shown in Figure 2d,e, the PL intensity of the main emission peak increased gradually when the water content changed from 0% to 60% but decreased from 60% to 90%. As shown in Figure 2f, the PL intensity reached a maximum when the volume fraction of water was 50%. We speculated that the reason for the attenuation of PL intensity shown in Figure 2b–f was similar to that in Figure 2a. The variation trend of the maximum fluorescence intensity of compounds **2**–**6** under different water contents is shown in Appendix A–f. In summary, compounds **2**–**6** were shown to possess AIEE characteristics from these results of the PL spectra. As shown in Appendix A–f, they also had a large Stokes shift of about 135, 127, 135, 197, and 212 nm, respectively. In addition, we investigated the fluorescence stability properties of compounds **1**–**6** by measuring their time-dependent fluorescence spectra. As is shown in Appendix A, with the passage of time, the intensity of the fluorescence spectra of all the compounds had no obvious change, which means that compounds **1**–**6** in the CH_3_OH/H_2_O mixed solution were stable. This feature provides strong support for their application in other fields. However, as shown in Appendix A–e, compounds **7–11** exhibited the strongest fluorescence in the pure CH_3_OH solution and showed an apparent decrease from 0–90%, which means compounds **7–11** are ACQ molecules [39].

In order to further study the AIEE properties of compounds **1**–**6**, the UV–vis absorption spectra of these compounds in the pure CH_3_OH solution and CH_3_OH/H_2_O mixed solution with 90% water contents were measured. The results are shown in Figure 3. In the pure CH_3_OH solution, the maximum absorption wavelengths of compounds **1**–**6** were 345, 341, 342, 341, 318, and 306 nm, respectively. As the water contents increased to 90%, the maximum absorption peaks of compounds **1**, **2**, **3**, **5** showed a significant red shift, and the absorption peaks of compounds **4** and **6** disappeared, indicating that new aggregated particles formed in the solution with the increase in undesirable solvent (water) [40].

The AIEE properties of compounds **1**–**6** can also be proved through their fluorescence photographs under a 365 nm wavelength UV light. As can be observed intuitively from Figure 4a, there was almost no fluorescence emission in the pure CH_3_OH solution. Interestingly, when the water content increased to 70%, a distinct bright green color was captured. However, the fluorescence emission decreased somewhat as the water content continued to increase to 90%. Similar results can be observed in Figure 4b–f. These results are consistent with the trend seen in the fluorescence spectra results above. In addition, we found that compounds **1**–**6** showed different color emissions, including green, blue, and yellow. This phenomenon can be attributed to the fact that they have different substituent groups, which affect the electronic density distribution of the molecule. This indicates that the fluorescence color of chalcone derivatives can be successfully adjusted and controlled by changing the substituent functional groups [41].

In order to further research the AIEE properties of compounds **1**–**6**, we measured their fluorescence quantum yields (ϕF) in mixed solutions of CH_3_OH/H_2_O with different water contents. As shown in Table 2, the ϕF of compound **1** in pure CH_3_OH solution was only 0.03. Whereas, when the water fraction was 70%, the ϕF increased to 0.59. Subsequently, when the water content continued to increase to 90%, the ϕF decreased slightly to 0.34. Obviously, the variation trend of the fluorescence quantum yield is consistent with the fluorescence spectra results. Similarly, the variation trend of the fluorescence quantum yield of compounds **2**–**6** in mixed solutions with different water contents are in accordance with the change trend of the fluorescence spectra. These results further validate that compounds **1**–**6** show typical AIEE properties.

In order to explain the AIEE characteristics of compound **1**–**6** more directly, the morphology of compounds **1**–**6** in the CH_3_OH/H_2_O mixed solutions with different water contents was observed by scanning electron microscopy (SEM). In Figure 5a, regular, spherical nanoparticles with small diameters can be directly observed. When the water content increased to 70%, the diameter of the spherical nanoparticles became larger, and they were still uniformly distributed in the solution, as shown in Figure 5b. However, as shown in Figure 5c, when the water content increased to 90%, a large number of amorphous particles gathered in the solution. This may be the reason for the decrease in fluorescence intensity when the water content was 90% [42]. The SEM results of compounds **2**–**6** also showed the same change, which indicates that the increase in the diameter of the spherical nanoparticles was conducive to the enhancement of fluorescence intensity. However, when amorphous particles appeared in the solution, the fluorescence emission reduced to a certain extent, which is consistent with the results of the fluorescence spectrum.

In our previous studies, we found that RIR should be the main cause for the AIEE properties of flavonoids and flavanones [24,25]. Therefore, we performed different viscosity tests to explore the AIEE mechanism of compounds **1**–**6**. By adding ethylene glycol (EG) to change the viscosity of the solution, we measured the fluorescence intensity of the compounds in the CH_3_OH/EG mixed solution with different EG fractions (0–50%). As shown in Figure 6a, when the EG fractions gradually increased from 0% to 50%, the viscosity of the solution gradually increased, as did the fluorescence intensity. The fluorescence intensity of compounds **2**–**6** in the mixed solutions with different viscosities also showed the same variation rule. According to the above experimental results, when the content of EG in the mixed solution increased, the viscosity of the solution gradually increased, which restricted the free rotation within the molecule and enhanced the fluorescence intensity of the mixed solution. Therefore, we speculated that the restriction of intramolecular rotation (RIR) might be the main reason for the AIEE characteristics of compounds **1**–**6**.

### 2.2. Cell Imaging

In order to evaluate the application of compounds **1**–**6** in the biomedical field, we used an MTT assay to test the cytotoxicity of compounds **1**–**6** with different concentrations (1, 3, 6, and 12 µM) on A549 cells, respectively. As shown in Figure 7 below, after 24 h co-incubation of A549 cells with compounds **1**–**6**, cell viability at various experimental concentrations was more than 80%, which indicates that compounds **1**–**6** have low cytotoxicity on and good cytocompatibility with A549 cells at the experimental concentrations. This result indicates the possibility of its practical application in the biomedical field.

The above experimental results indicate that compounds **1**–**6** are typical AIEE molecules with good photostability, low cytotoxicity, and good biocompatibility. Therefore, in order to further study the biological applications of compounds **1**–**6**, cell uptake experiments were carried out. A549 cells were co-cultured with 10 µM compounds **1**–**6** in a cell incubator in darkness for 30 min and then washed with PBS buffer three times for confocal imaging. All fluorescence images results are shown in Figure 8. After co-incubating with compounds **1**–**6** for 30 min, the cell morphology of A549 cells did not change significantly, and a strong blue fluorescence was observed in the cytoplasm. These results indicate that compounds **1**–**6** could be taken up by A549 cells and aggregated in the cytoplasm, which again verified that compounds **1**–**6** had good cytocompatibility. In addition, this means that compounds **1**–**6** have great application potential in the field of biological imaging.

From the cell uptake experiment results above, we can intuitively observe the strong blue fluorescence in the cytoplasm. Based on our previous studies, both flavonoids and flavanones were specifically aggregated in the mitochondria [24,25]. In order to further study whether chalcones also aggregated in the mitochondria, we conducted a mitochondrial co-localization experiment. A549 cells were co-cultured with compounds **1**–**6** (10 µM) and Mito Tracker Deep Red (200 nM) for 30 min in the dark, respectively. After washing three times with PBS buffer, the cells were observed by FV3000 laser scanning confocal microscope (Zeiss) under a 60-fold oil immersion lens. The experimental results are shown in Figure 9. Figure 9a,d,g,j,m,p, with obvious blue fluorescence, are the cell images of compounds **1**–**6**. In Figure 9b,e,h,k,n,q, MT specifically labels the target organelle mitochondria, and we can see strong red fluorescence in the cytoplasm. In the merged Figure 9c,f,i,l,o,r, it can be seen that the blue fluorescence region emitted by the compounds is highly coincidental with the red fluorescence region emitted by the MT. In order to further accurately analyze the coincidence, cellSens software was used for quantitative analysis, and the results are shown in Table 3 below. The corresponding R values (Pearson correlation coefficient, the coefficient indicating the degree of coincidence, range +1 to −1) of compounds **1**–**6** were 0.87, 0.83, 0.87, 0.88, 0.86, and 0.85, respectively, which indicate a high degree of overlap between the compounds and the MT staining regions. The results of the mitochondrial co-localization experiment showed that compounds **1**–**6** specifically aggregated in the mitochondria, providing strong support for their practical application in mitochondrial imaging.

The above experimental results indicate that compounds **1**–**6** can be used in the field of cell imaging. In order to further study their application in animal imaging, zebrafish were selected as the study object. The advantages of zebrafish as a model organism are outstanding. They have a small body, a short development cycle, and a high level (about 87%) of genetic homology with humans. Moreover, the living embryos are transparent, so we can directly observe the distribution of drugs in zebrafish embryos [43,44,45]. As can be seen from Figure 10, 24 h after administration, blue fluorescence was observed in the yolk of zebrafish embryos. This means that compounds **1**–**6** entered the embryo through the embryo membrane within 1 h and mainly concentrated in the yolk sac. After 48 h, we could observe distinct blue fluorescence at the same location in the zebrafish embryos. After 120 h, the zebrafish embryos had developed into normal young fish, and strong blue fluorescence was distributed in the yolk sac and digestive tract of the zebrafish. The above results indicate that AIEE molecules **1**–**6** can be applied in in vivo imaging and specifically label the digestive tract of zebrafish.

## 3. Materials and Methods

### 3.1. Materials and Instruments

All the regents and analytical grade solvents in this paper were acquired from reagent providers and used without further purification unless otherwise indicated. 2-methoxybenzaldehyde, 3-methoxybenzaldehyde, 4-methoxybenzaldehyde, 4-methoxyacetophenone, 4-methylacetophenone, 4-dimethylaminobenzaldehyde, 4-methylbenzaldehyde, and 3-methylbenzaldehyde were purchased from Aladdin (Shanghai, China). Acetophenone and benzaldehyde were obtained from Macklin (Shanghai, China). Sodium hydroxide (NaOH), ethanol, methanol, and glycol solutions were purchased from Zhiyuan Chemical Reagent Co. Ltd. (Tianjin, China). The high-resolution mass spectra were measured on a Orbitrap Fusion Lumos mass spectrometers (ThermoFisher, Waltham, MA, America). All the consumables and raw materials used in the cell experiment, as well as the related instruments and equipment used throughout the research, can be obtained from reference [25].

### 3.2. Synthesis of Chalcones Derivatives

The general synthetic procedure for the chalcone derivatives is shown in Figure 1b. This synthesis has only one step. Under the catalysis of 10% sodium hydroxide, acetophenone and benzaldehyde containing different substitutions undergo the Claisen–Schmidt reaction to produce the corresponding products [46]. During the reaction process, TLC with 10% ethyl acetate/petroleum ether was used as the solvent system to detect the reaction progress until all reactants disappeared. The TLC Rf values of compounds **1**–**11** are 0.41, 0.31, 0.22, 0.36, 0.24, 0.42, 0.53, 0.28, 0.53, 0.59, 0.36, respectively. We recrystallized the solids using 95% ethanol to obtain the pure end products. Fortunately, the yields of all the end products was relatively high, with all above 75%. The ^1^H, ^13^C-NMR spectra and ESI-MS analysis for compounds **1**–**11** can be obtained from the Appendix A.

#### 3.2.1. Synthesis of Compound **1**

Sodium hydroxide (0.55 g) and water (5 mL) were charged into a 50 mL flask, and then we added a mixture of acetophenone (1.2 mL) and ethanol (3 mL). All reagents were dissolved under a magnetic stirrer. Under this condition, 2-methoxybenzaldehyde (1.36 g) was dissolved in 1 mL ethanol and was added slowly through the dripping funnel. We adjusted the drip rate to maintain the reaction temperature between 25 and 30 °C. After dripping 2-methoxybenzaldehyde, the mixed solution was vigorously stirred at room temperature for about 5 h until there was solid precipitation. Then, we placed the bottle in an ice bath and cooled for 15–30 min until the crystal was completely separated. Then, the white precipitation was collected by vacuum filtration and washed with distilled water until the litmus paper was neutral. Finally, the crude product was recrystallized with 95% ethanol to generate a pure final 2-methoxychalcone product.

#### 3.2.2. Synthesis of Compound **2**

Sodium hydroxide (0.55 g) and water (5 mL) were charged into a 50 mL flask, and then we added a mixture of acetophenone (1.2 mL) and ethanol (3 mL). All reagents were dissolved under a magnetic stirrer. Under this condition, 4-methoxybenzaldehyde (1.2 mL) was dissolved in 1 mL ethanol and was added slowly through the dripping funnel. The latter operation was the same as that in Section 3.2.1. Finally, we obtained the target 4-methoxychalcone product.

#### 3.2.3. Synthesis of Compound **3**

Sodium hydroxide (0.55 g) and water (5 mL) were charged into a 50 mL flask, and then, we added a mixture of 4-methoxyacetophenone (1.65 g) and ethanol (3 mL). All reagents were dissolved under a magnetic stirrer. Under this condition, 4-methoxybenzaldehyde (1.2 mL) was dissolved in 1 mL ethanol and was added slowly through the dripping funnel. The latter operation was the same as that in Section 3.2.1. Finally, we obtained the target 4-methoxy-4′-methoxychalcone product.

#### 3.2.4. Synthesis of Compound **4**

Sodium hydroxide (0.55 g) and water (5 mL) were charged into a 50 mL flask, and then we added a mixture of 4-methylacetophenone (1.46 mL) and ethanol (3 mL). All reagents were dissolved under a magnetic stirrer. Under this condition, 4-methoxybenzaldehyde (1.2 mL) was dissolved in 1 mL ethanol and was added slowly through the dripping funnel. The latter operation was the same as that in Section 3.2.1. Finally, we obtained the target 4-methoxy-4′-methyl chalcone product.

#### 3.2.5. Synthesis of Compound **5**

Sodium hydroxide (0.55 g) and water (5 mL) were charged into a 50 mL flask, and then we added the mixture of 4-methoxyacetophenone (1.65 g) and ethanol (3 mL). All reagents were dissolved under a magnetic stirrer. Under this condition, 3-methoxybenzaldehyde (1.2 mL) was dissolved in 1 mL ethanol and was added slowly through the dripping funnel. The latter operation was the same as that in Section 3.2.1. Finally, we obtained the target 3-methoxy-4′-methoxychalcone product.

#### 3.2.6. Synthesis of Compound **6**

Sodium hydroxide (0.55 g) and water (5 mL) were charged into a 50 mL flask, and then we added a mixture of 4-methylacetophenone (1.46 mL) and ethanol (3 mL). All reagents were dissolved under a magnetic stirrer. Under this condition, 3-methoxybenzaldehyde (1.2 mL) was dissolved in 1 mL ethanol and was added slowly through the dripping funnel. The latter operation was the same as that in Section 3.2.1. Finally, we obtained the target 3-methoxy-4′-methyl chalcone product.

#### 3.2.7. Synthesis of Compound **7**

Sodium hydroxide (0.55 g) and water (5 mL) were charged into a 50 mL flask, and then we added a mixture of acetophenone (1.2 mL) and ethanol (3 mL). All reagents were dissolved under a magnetic stirrer. Under this condition, benzaldehyde (1 mL) was dissolved in 1 mL ethanol and was added slowly through the dripping funnel. The latter operation was the same as that in Section 3.2.1. Finally, we obtained the target chalcone product.

#### 3.2.8. Synthesis of Compound **8**

Sodium hydroxide (0.55 g) and water (5 mL) were charged into a 50 mL flask, and then we added a mixture of acetophenone (1.2 mL) and ethanol (3 mL). All reagents were dissolved under a magnetic stirrer. Under this condition, 4-dimethylaminobenzaldehyde (1.5 g) was dissolved in 1 mL ethanol and was added slowly through the dripping funnel. The latter operation was the same as that in Section 3.2.1. Finally, we obtained the target 4-dimethylaminochalcone product.

#### 3.2.9. Synthesis of Compound **9**

Sodium hydroxide (0.55 g) and water (5 mL) were charged into a 50 mL flask, and then we added a mixture of acetophenone (1.2 mL) and ethanol (3 mL). All reagents were dissolved under a magnetic stirrer. Under this condition, 4-methylbenzaldehyde (1.2 mL) was dissolved in 1 mL ethanol and was added slowly through the dripping funnel. The latter operation was the same as that in Section 3.2.1. Finally, we obtained the target 4-methyl chalcone product.

#### 3.2.10. Synthesis of Compound **10**

Sodium hydroxide (0.55 g) and water (5 mL) were charged into a 50 mL flask, and then we added a mixture of acetophenone (1.2 mL) and ethanol (3 mL). All reagents were dissolved under a magnetic stirrer. Under this condition, 3-methylbenzaldehyde (1.2 mL) was dissolved in 1 mL ethanol and was added slowly through the dripping funnel. The latter operation was the same as that in Section 3.2.1. Finally, we obtained the target 3-methyl chalcone product.

#### 3.2.11. Synthesis of Compound **11**

Sodium hydroxide (0.55 g) and water (5 mL) were charged into a 50 mL flask, and then we added the mixture of 4-methoxyacetophenone (1.5 g) and ethanol (3 mL). All reagents were dissolved under a magnetic stirrer. Under this condition, benzaldehyde (1 mL) was dissolved in 1 mL ethanol and was added slowly through the dripping funnel. The latter operation was the same as that in Section 3.2.1. Finally, we obtained the target 4′-methoxychalcone product.

#### 3.2.12. Characterization of Compounds **1**–**11**

2-Methoxychalcone (**1**), yield: 89%; slight yellow solid, m.p.: 59–60 °C; ^1^H NMR (500 MHz, CDCl_3_) δ 8.12 (d, *J* = 15.9 Hz, 1H), 8.06–7.98 (m, 2H), 7.66–7.64 (m, 1H), 7.62 (d, *J* = 12.4 Hz, 1H), 7.58 (t, *J* = 7.4 Hz, 1H), 7.50 (t, *J* = 7.5 Hz, 2H), 7.43–7.34 (m, 1H), 7.00 (t, *J* = 7.5 Hz, 1H), 6.95 (d, *J* = 8.3 Hz, 1H), 3.92 (s, 3H); ^13^C NMR (126 MHz, CDCl_3_) δ 191.17, 158.82, 140.43, 138.53, 132.54, 131.76, 129.25, 128.54, 123.94, 122.89, 120.75, 111.24, 55.55; HRMS (ESI-MS) *m*/*z* calcd for C_16_H_15_O_2_ [M + H]^+^: 239.10666; found: 239.10677.

4-Methoxychalcone (**2**), yield: 92%; slight yellow solid, m.p.: 75–76 °C; ^1^H NMR (500 MHz, CDCl_3_) δ 8.03–7.99 (m, 2H), 7.79 (d, *J* = 15.6 Hz, 1H), 7.62–7.55 (m, 3H), 7.50 (t, *J* = 7.6 Hz, 2H), 7.42 (d, *J* = 15.6 Hz, 1H), 6.96–6.92 (m, 2H), 3.85 (s, 3H); ^13^C NMR (126 MHz, CDCl_3_) δ 190.62, 161.69, 144.73, 138.50, 132.57, 130.24, 128.57, 128.42, 127.61, 119.78, 114.43, 55.42; HRMS (ESI-MS) *m*/*z* calcd for C_16_H_15_O_2_ [M + H]^+^: 239.10666; found: 239.10674.

4-Methoxy-4′-methoxychalcone (**3**), yield: 75%; slight yellow solid, m.p.: 101–102 °C; 1H NMR (500 MHz, CDCl3) δ 8.03 (d, *J* = 8.7 Hz, 2H), 7.78 (d, *J* = 15.6 Hz, 1H), 7.60 (d, *J* = 8.6 Hz, 2H), 7.43 (d, *J* = 15.6 Hz, 1H), 6.98 (d, *J* = 8.7 Hz, 2H), 6.94 (d, *J* = 8.6 Hz, 2H), 3.89 (s, 3H), 3.86 (s, 3H); 13C NMR (126 MHz, CDCl3) δ 188.75, 163.27, 161.51, 143.80, 131.35, 130.70, 130.10, 127.81, 119.54, 114.38, 113.78, 55.47, 55.39; HRMS (ESI-MS) *m*/*z* calcd for C_17_H_17_O_3_ [M + H]^+^: 269.11722; found: 269.11728.

4-Methoxy-4′-methylchalcone (**4**), yield: 78%; slight yellow solid, m.p.: 90–93 °C; ^1^H NMR (500 MHz, CDCl_3_) δ 7.93 (d, *J* = 8.1 Hz, 2H), 7.78 (d, *J* = 15.6 Hz, 1H), 7.60 (d, *J* = 8.7 Hz, 2H), 7.42 (d, *J* = 15.6 Hz, 1H), 7.30 (d, *J* = 8.0 Hz, 2H), 6.94 (d, *J* = 8.7 Hz, 2H), 3.86 (s, 3H), 2.43 (s, 3H); ^13^C NMR (126 MHz, CDCl_3_) δ 190.06, 161.59, 144.25, 143.37, 135.90, 130.17, 129.27, 128.57, 127.73, 119.78, 114.40, 55.40, 21.66; HRMS (ESI-MS) *m*/*z* calcd for C_17_H_17_O_2_ [M + H]^+^: 253.12231; found: 253.12238.

3-Methoxy-4′-methoxychalcone (**5**), yield: 83%; slight yellow solid, m.p.: 103–104 °C; ^1^H NMR (500 MHz, CDCl_3_) δ 8.04 (d, *J* = 8.8 Hz, 2H), 7.76 (d, *J* = 15.6 Hz, 1H), 7.52 (d, *J* = 15.6 Hz, 1H), 7.33 (t, *J* = 7.9 Hz, 1H), 7.24 (d, *J* = 7.7 Hz, 1H), 7.16 (s, 1H), 6.99 (d, *J* = 8.8 Hz, 2H), 6.96 (dd, *J* = 8.2, 2.4 Hz, 1H), 3.89 (s, 3H), 3.86 (s, 3H); ^13^C NMR (126 MHz, CDCl_3_) δ 188.72, 163.45, 159.93, 143.88, 136.47, 131.06, 130.84, 129.91, 122.19, 121.00, 116.06, 113.86, 113.41, 55.50, 55.35; HRMS (ESI-MS) *m*/*z* calcd for C_17_H_17_O_3_ [M + H]^+^: 269.11722; found: 269.11723.

3-Methoxy-4′-methyl chalcone (**6**), yield: 81%; slight yellow solid, m.p.: 65–67 °C; ^1^H NMR (500 MHz, CDCl_3_) δ 7.93 (d, *J* = 8.1 Hz, 2H), 7.76 (d, *J* = 15.7 Hz, 1H), 7.51 (d, *J* = 15.7 Hz, 1H), 7.34 (d, *J* = 7.9 Hz, 1H), 7.31 (d, *J* = 8.1 Hz, 2H), 7.24 (d, *J* = 7.7 Hz, 1H), 7.16 (s, 1H), 6.96 (dd, *J* = 8.2, 2.3 Hz, 1H), 3.86 (s, 3H), 2.44 (s, 3H); ^13^C NMR (126 MHz, CDCl_3_) δ 190.03, 159.94, 144.31, 143.67, 136.38, 135.60, 129.93, 129.34, 128.67, 122.42, 121.05, 116.17, 113.43, 55.35, 21.68; HRMS (ESI-MS) *m*/*z* calcd for C_17_H_17_O_2_ [M + H]^+^: 253.12231; found: 253.12239.

Chalcone (**7**), yield: 89%; slight yellow solid, m.p.: 59 °C; ^1^H NMR (500 MHz, CDCl_3_) δ 8.07–7.97 (m, 2H), 7.82 (d, *J* = 15.7 Hz, 1H), 7.65 (dd, *J* = 6.1, 2.8 Hz, 2H), 7.61–7.57 (m, 1H), 7.54 (d, *J* = 14.3 Hz, 1H), 7.53–7.48 (m, 2H), 7.47–7.36 (m, 3H); ^13^C NMR (101 MHz, CDCl_3_) δ 189.52, 143.81, 137.16, 133.83, 131.76, 129.52, 127.93, 127.60, 127.47, 127.42, 121.04; HRMS (ESI-MS) *m*/*z* calcd for C_15_H_13_O [M + H]^+^: 209.09609; found: 209.09619.

4-Dimethylaminochalcone (**8**), yield: 76%; bright orange solid, m.p.: 113–114 °C; ^1^H NMR (500 MHz, CDCl_3_) δ 8.00 (d, *J* = 6.9 Hz, 2H), 7.79 (d, *J* = 15.5 Hz, 1H), 7.57 (dd, *J* = 15.3, 7.7 Hz, 3H), 7.49 (t, *J* = 7.1 Hz, 2H), 7.37 (d, *J* = 15.5 Hz, 1H), 7.02–6.69 (m, 2H), 3.07 (s, 6H); ^13^C NMR (126 MHz, CDCl_3_) δ 190.71, 151.95, 145.83, 139.06, 132.15, 130.41, 128.45, 128.31, 122.77, 117.00, 111.93, 40.19; HRMS (ESI-MS) *m*/*z* calcd for C_17_H_18_ON [M + H]^+^: 252.13829; found: 252.13837.

4-Methyl chalcone (**9**), yield: 85%; slight yellow solid, m.p.: 93 °C; ^1^H NMR (500 MHz, CDCl_3_) δ 8.05–7.98 (m, 2H), 7.80 (d, *J* = 15.7 Hz, 1H), 7.58 (t, *J* = 7.4 Hz, 1H), 7.55 (d, *J* = 8.1 Hz, 2H), 7.53–7.50 (m, 2H), 7.49 (d, *J* = 4.6 Hz, 1H), 7.23 (d, *J* = 7.9 Hz, 2H), 2.40 (s, 3H); ^13^C NMR (101 MHz, CDCl_3_) δ 189.64, 143.93, 140.07, 137.32, 131.65, 131.11, 128.68, 127.56, 127.46, 127.44, 120.05, 20.51; HRMS (ESI-MS) *m*/*z* calcd for C_16_H_15_O [M + H]^+^: 223.11174; found: 223.11177.

3-Methyl chalcone (**10**), yield: 81%; slight yellow solid, m.p.: 63 °C; ^1^H NMR (500 MHz, CDCl_3_) δ 8.05–8.01 (m, 2H), 7.79 (d, *J* = 15.7 Hz, 1H), 7.62–7.56 (m, 1H), 7.53 (d, *J* = 7.6 Hz, 1H), 7.50 (d, *J* = 7.1 Hz, 2H), 7.46 (d, *J* = 7.4 Hz, 2H), 7.32 (t, *J* = 7.9 Hz, 1H), 7.24 (d, *J* = 7.6 Hz, 1H), 2.41 (s, 3H); ^13^C NMR (101 MHz, CDCl_3_) δ 189.58, 144.05, 137.60, 137.23, 133.80, 131.71, 130.40, 128.02, 127.82, 127.58, 127.47, 124.69, 120.85, 20.32; HRMS (ESI-MS) *m*/*z* calcd for C_16_H_15_O [M + H]^+^: 223.11174; found: 223.11178.

4′-Methoxychalcone (**11**), yield: 88%; slight yellow solid, m.p.: 96 °C; ^1^H NMR (500 MHz, CDCl3) δ 8.05 (d, *J* = 8.9 Hz, 2H), 7.81 (d, *J* = 15.7 Hz, 1H), 7.67–7.62 (m, 2H), 7.55 (d, *J* = 15.7 Hz, 1H), 7.45–7.39 (m, 3H), 6.99 (d, *J* = 8.9 Hz, 2H), 3.89 (s, 3H); ^13^C NMR (101 MHz, CDCl3) δ 187.69, 162.40, 142.94, 134.04, 130.04, 129.79, 129.31, 127.89, 127.33, 120.83, 112.82, 54.46; HRMS (ESI-MS) *m*/*z* calcd for C_16_H_15_O_2_ [M + H]^+^: 239.10666; found: 239.10677.

### 3.3. Preparatory Work before UV–Vis Spectra, PL Spectra, and SEM Measurements

#### 3.3.1. Preparatory Work before UV–Vis Spectra and PL Spectra Measurements

Compounds **1**–**11** were weighed accurately using electronic scales: 0.50, 0.50, 0.56, 0.53, 0.56, 0.53, 0.44, 0.53, 0.47, 0.47, and 0.50 mg. Then, they were each dissolved in a methanol solution (10 mL) and mixed well. The final concentration of each solution was 2.07 × 10^−4^ M. Then, each mother solution was diluted with different amounts of water and methanol. Finally, the solutions with a water fraction from 0 to 90 vol % were prepared at a concentration of 2.07 × 10^−5^ M. After being sonicated for 15 min, the solutions were measured immediately at room temperature to obtain the UV-vis absorption spectra and the PL spectra at their excitation wavelengths of 350, 345, 348, 345, 360, 363, 313, 429, 326, 318, and 318 nm, respectively. The sample preparation method for the fluorescence quantum yield (ϕF) test of compounds **1**–**6** was conducted in the same manner as described above.

#### 3.3.2. Preparatory Work before SEM Measurements

All samples were configured in the same way as described in Section 3.3.1. Compounds **1** and **2** were prepared in a CH_3_OH/H_2_O mixture with different water fractions of 50%, 70%, and 90%, respectively. Compound **3** was prepared in a CH_3_OH/H_2_O mixture with different water fractions of 50%, 80% and 90%, respectively. Compounds **4**, **5**, and **6** were prepared in a CH_3_OH/H_2_O mixture with different water fractions of 50%, 60%, and 90%, respectively. After sonication for 15 min, the mixed solutions were added carefully to silicon wafers, separately. After being left at room temperature for 24 h, the silicon wafers had evaporated and were tested by scanning electron microscopy (SEM) afterwards.

### 3.4. Preparatory Work before Ethylene Glycol (EG) Measurements

Compounds **1**–**6** were dissolved in a methanol solution, respectively, and mixed well with a final concentration of 2.07 × 10^−4^ M. Then, we diluted the solution with different EG fractions (fw = 0%, 10%, 20%, 30%, 40%, 50%) of EG/CH_3_OH mixed solutions to ensure a final concentration of 2.07 × 10^−5^ M. The above solution was sonicated for 15 min. Then, we immediately measured their PL spectra at room temperature with excitation wavelengths of 350, 345, 348, 345, 360, and 363 nm, respectively.

### 3.5. Cell Culture

RPMI-1640 culture medium containing 10% fetal bovine serum (FBS), 10 mg/mL streptomycin, and 10 mg/mL penicillin was used for culturing the A549 cells in the cell culture incubator with 5% CO_2_ and 90% humidity at 37 °C.

### 3.6. Evaluation of Cell Viability

A549 cells in the logarithmic growth phase were transferred to 96-well plates, and there were about 9 × 10^3^ cells in each well. The 96-well plates were placed in an incubator for 24 h. Then, we transferred the preconfigured sample solutions of compounds **1**–**6** of different concentrations (1, 3, 6, 12 µM) to each well, respectively After culturing the cells in an incubator for 24 h, we added 20 μL 0.5% MTT solution to each well. After incubation for 4 h, the excess solution in each well was sucked out carefully. Then, 150 μL DMSO solution was added to each well and shaken through a shaker for 10 min until the crystals were completely dissolved. The absorbance value at 490 nm of the plates was measured by a microplate reader (Synergy H1).

### 3.7. Cell Imaging

A549 cells in the logarithmic growth phase were transferred to glass bottom dishes with a density in each dish of 1 × 10^5^. The cells grew in the cell culture incubator for 24 h. Then, we added the sample solution of compounds **1**–**6** with a concentration of 10 μM to each dish, respectively. After 30 min of dark incubation, the solution in the dishes was carefully sucked out with pipetting gun and washed with PBS buffer for three times. Finally, we added 1 mL PBS buffer to each dish and observed them under a 60-fold oil immersion lens using an FV3000 laser scanning confocal microscope (Zeiss). The procedure of mitochondrial colocalization imaging was similar to the above procedure. After the A549 cells were transferred to glass bottom dishes and cultivated in cell incubators for 24 h, the mixed solutions of compounds **1**–**6** (10 µM) and Mito Tracker Red (MT) (200 nM) were added, respectively. After 30 min of dark incubation in the incubator, the solution was sucked out carefully, and the dishes were washed three times by PBS buffer. Finally, 1 mL PBS buffer was added to each dish, and they were observed by an FV3000 laser scanning confocal microscope (Zeiss) under a 60-fold oil immersion lens.

### 3.8. Zebrafish Imaging

Zebrafish (wild-type AB strain) were purchased from the Zebrafish Resource Center (Core Lab Plat for Medical Science, Zhongshan School of Medicine, Sun Yat-sen University). All procedures were conducted in accordance with the Code of Ethics for Animal Experiments. The ethical approval number is SYSU-IACUC-2020-B0659. The zebrafish were fed at a constant temperature of 28.5 °C and given a light cycle of 14 h on/10 h off. Before the fluorescence imaging experiment, 6 hpf zebrafish embryos were randomly divided into six groups according to the random grouping method and given a sample solution of compounds **1**–**6** with a concentration of 15 µM. After incubating at a constant temperature of 28.5 °C for 1 h, the excess liquid was discarded, and we washed the embryos three times with E3 medium. Then, the embryos were incubated continuously in an incubator with a constant temperature of 28.5 °C. After 24 h and 48 h, we observed the embryos using an EVOS FL Auto Imaging system. After 120 h, we observed the Zebrafish using FV3000 laser scanning confocal microscopy under a 10-fold lens.

## 4. Conclusions

In summary, we successfully designed and synthesized chalcone derivatives **1**–**11** based on our previous research of AIEE molecules. By analyzing the experimental results of PL intensity, fluorescence photographs under a 365 nm wavelength UV light and fluorescence quantum yields (ϕF), compounds **1**–**6** were identified as typical AIEE molecules. Through different viscosity tests, we determined that the main reason for the AIEE phenomenon was the restriction of intramolecular rotation (RIR). Furthermore, compounds **1**–**6** showed low cytotoxicity and good cytocompatibility at the experimental concentrations. Additionally, we found that they could specifically aggregate in the mitochondria, which enriches the compounds library that can be applied to mitochondrial imaging. We obtained pictures of compounds **1**–**6** in zebrafish fluorescence imaging from the experiment, which demonstrated that these AIEE molecules could be targeted to aggregate in the digestive tract and have the potential to be applied in in vivo imaging of zebrafish. Our research group is dedicated to the design and synthesis of additional novel AIEE molecules. Further efforts to study their application prospects in the field of living cell and biological imaging are in progress.

## Figures and Tables

**Figure 1 ijms-22-08949-f001:**
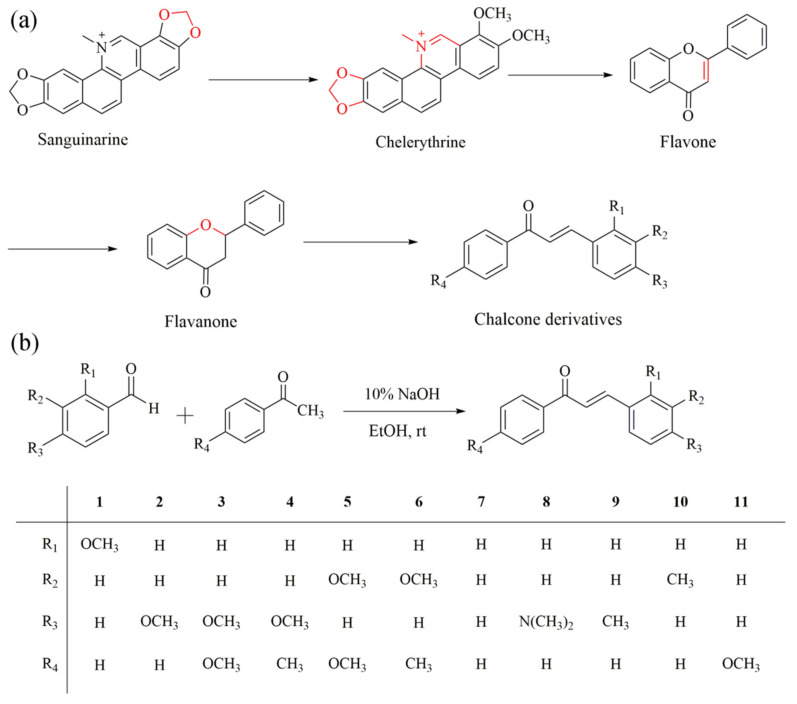
(**a**) Chemical structures of sanguinarine, chelerythrine, flavone, flavanone, and chalcone derivatives. (**b**) Synthesis route and chemical structure of chalcone derivatives **1**–**11**.

**Figure 2 ijms-22-08949-f002:**
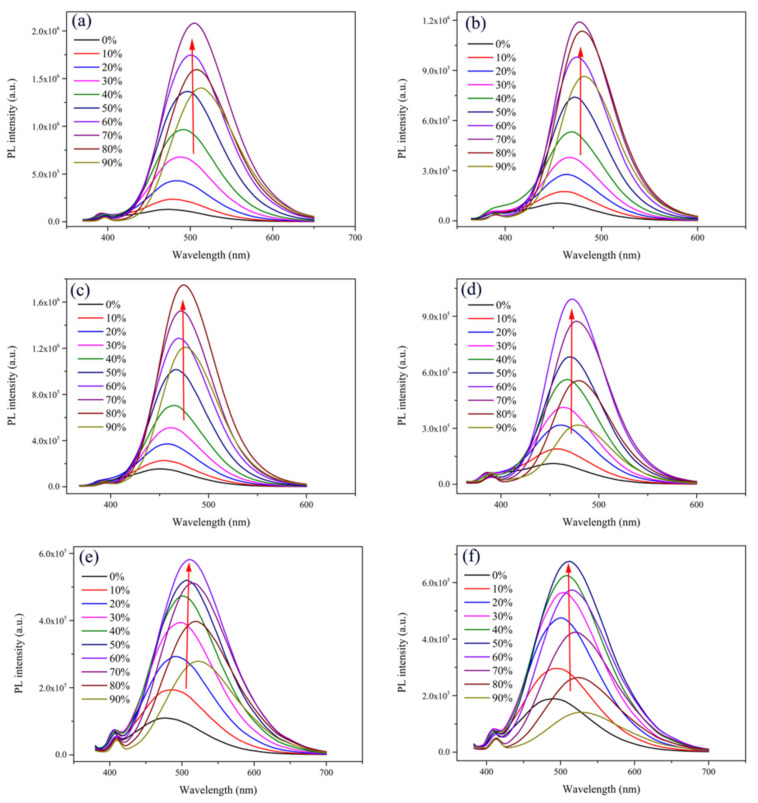
PL spectra of compounds (**a**) **1**, (**b**) **2**, (**c**) **3**, (**d**) **4**, (**e**) **5**, (**f**) **6** in CH_3_OH/H_2_O mixed solutions (c = 2.07 × 10^−5^ M) with different water fractions (0–90%).

**Figure 3 ijms-22-08949-f003:**
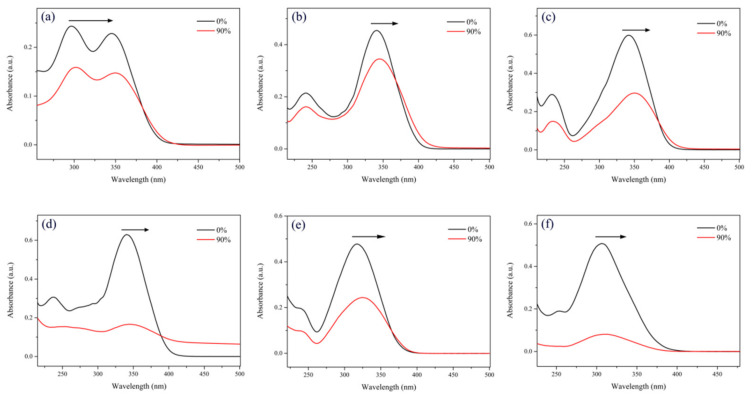
UV–vis absorption spectra of compounds (**a**) **1**, (**b**) **2**, (**c**) **3**, (**d**) **4**, (**e**) **5**, and (**f**) **6** in pure CH_3_OH solution and CH_3_OH/H_2_O mixed solution with 90% water contents.

**Figure 4 ijms-22-08949-f004:**
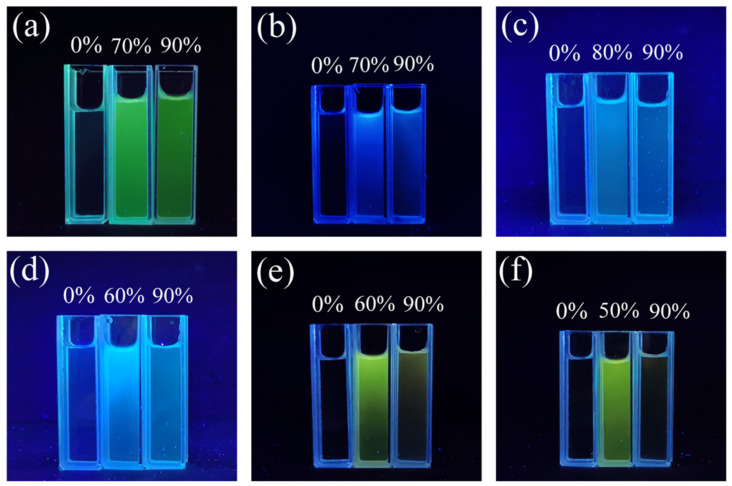
Fluorescence photographs of compounds (**a**) **1**, (**b**) **2**, (**c**) **3**, (**d**) **4**, (**e**) **5**, (**f**) **6** in CH_3_OH/H_2_O mixed solution (with various water fractions) under 365 nm wavelength UV light.

**Figure 5 ijms-22-08949-f005:**
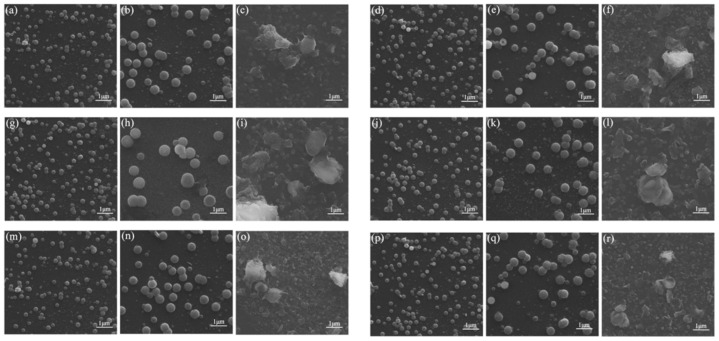
SEM images in CH_3_OH/H_2_O (5:5 **v**:**v**) mixed solutions of compounds (**a**) **1**, (**d**) **2**, (**g**) **3**, (**j**) **4**, (**m**) **5**, (**p**) **6**. SEM images in CH_3_OH/H_2_O (3:7 **v**:**v**) mixed solutions of compounds (**b**) **1**, (**e**) **2**. SEM image in CH_3_OH/H_2_O (2:8 **v**:**v**) mixed solutions of compound (**h**) **3**. SEM images in CH_3_OH/H_2_O (4:6 **v**:**v**) mixed solutions of compounds (**k**) **4**, (**n**) **5**, (**q**) **6**. SEM images in CH_3_OH/H_2_O (1:9 **v**:**v**) mixed solutions of compounds (**c**) **1**, (**f**) **2**, (**i**) **3**, (**l**) **4**, (**o**) **5**, (**r**) **6**.

**Figure 6 ijms-22-08949-f006:**
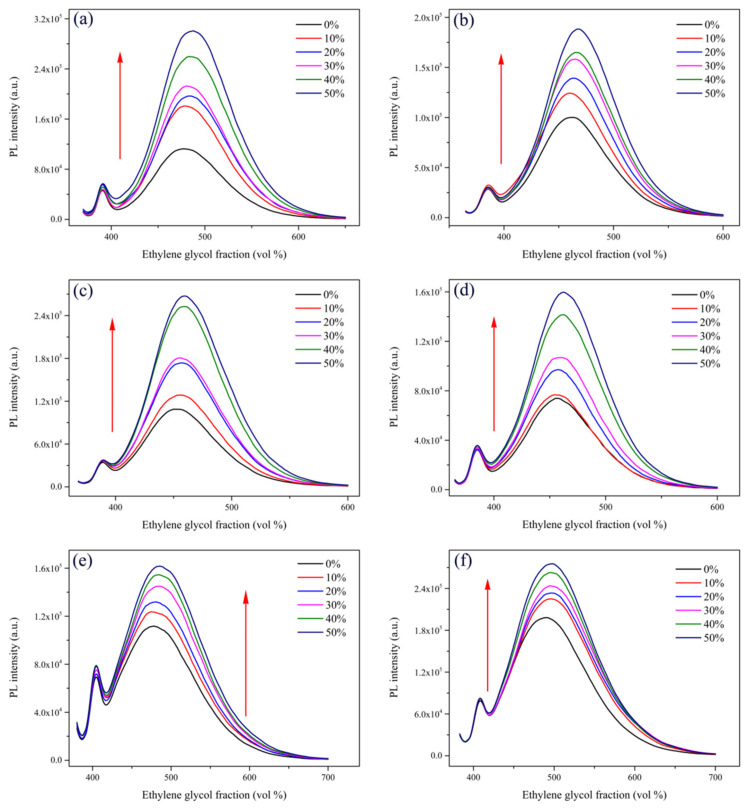
Fluorescence intensity of compounds (**a**) **1**, (**b**) **2**, (**c**) **3**, (**d**) **4**, (**e**) **5**, and (**f**) **6** in CH_3_OH/EG mixed solutions with different EG fractions (0–50%).

**Figure 7 ijms-22-08949-f007:**
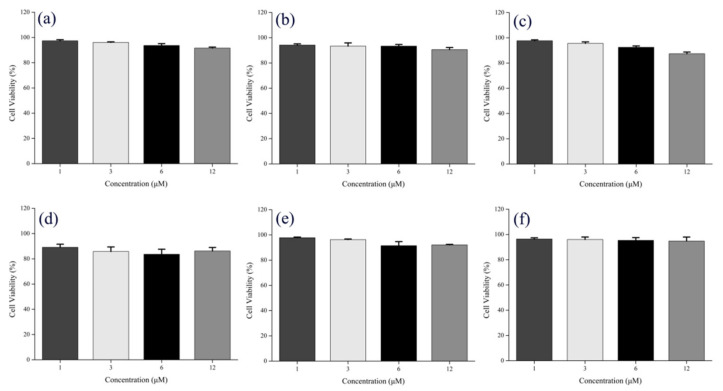
Cell viabilities of A549 cells co-cultured with different concentrations of compounds (**a**) **1**, (**b**) **2**, (**c**) **3**, (**d**) **4**, (**e**) **5**, (**f**) **6** for 24 h.

**Figure 8 ijms-22-08949-f008:**
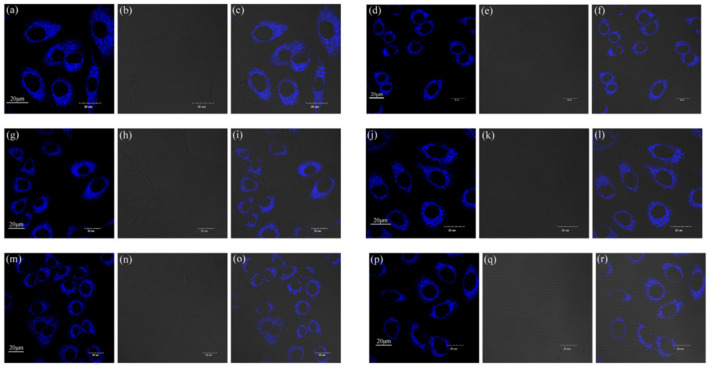
The imaging figures of A549 cells co-cultured with compounds (**a**) **1**, (**d**) **2**, (**g**) **3**, (**j**) **4**, (**m**) **5**, (**p**), **6** (10 µM) for 30 min. The bright-field images of A549 cells co-cultured with (**b**) **1**, (**e**) **2**, (**h**) **3**, (**k**) **4**, (**n**) **5**, (**q**) **6**. The merged images co-cultured with (**c**) **1**, (**f**) **2**, (**i**) **3**, (**l**) **4**, (**o**) **5**, (**r**) **6**. An excitation wavelength of 405 nm was selected for the experiment.

**Figure 9 ijms-22-08949-f009:**
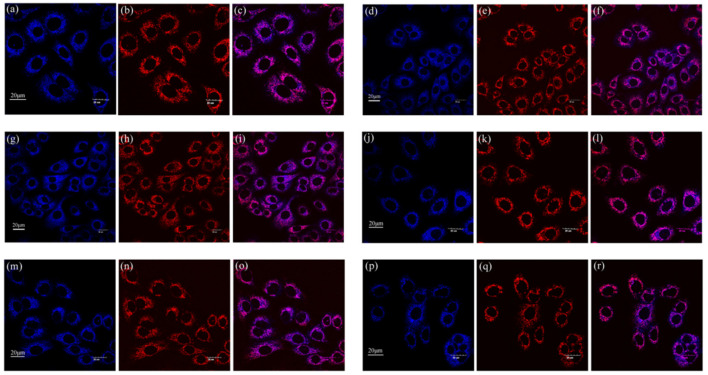
Co-localized images of A549 cells incubated with compounds (**a**) **1**, (**d**) **2**, (**g**) **3**, (**j**) **4**, (**m**) **5,** and (**p**) **6** (10 µM) and Mito Tracker Red (200 nM) (**b**) **1**, (**e**) **2**, (**h**) **3**, (**k**) **4**, (**n**) **5**, and (**q**) **6** for 30 min. The merged images co-cultured with (**c**) **1**, (**f**) **2**, (**i**) **3**, (**l**) **4**, (**o**) **5**, (**r**) **6**. The excitation wavelength of compounds **1**–**6** was 405 nm. The excitation wavelength of Mito Tracker Red was 578 nm.

**Figure 10 ijms-22-08949-f010:**
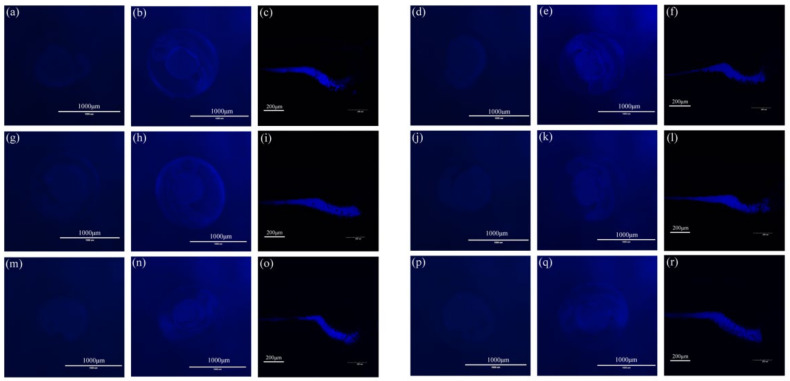
Fluorescence images of zebrafish embryos after soaking with compounds (**a**) **1**, (**d**) **2**, (**g**) **3**, (**j**) **4**, (**m**) **5,** and (**p**) **6** (15 µM) for 1 h at 24 hpf. Fluorescence images of zebrafish embryos after soaking with compounds (**b**) **1**, (**e**) **2**, (**h**) **3**, (**k**) **4**, (**n**) **5**, (**q**) **6** (15 µM) for 1 h at 48 hpf. Fluorescence images of zebrafish embryos after soaking with compounds (**c**) **1**, (**f**) **2**, (**i**) **3**, (**l**) **4**, (**o**) **5**, (**r**) **6** (15 µM) for 1 h at 120 hpf.

**Table 1 ijms-22-08949-t001:** MW, logp, and nON of compounds **1**–**11**.

Compound	MW	Logp	nON
**1**	239.11	3.46	2
**2**	239.11	3.46	2
**3**	269.12	3.33	3
**4**	253.12	3.95	2
**5**	269.12	3.33	3
**6**	253.12	3.95	2
**7**	209.10	3.59	1
**8**	252.14	3.87	2
**9**	223.11	4.07	1
**10**	223.11	4.07	1
**11**	239.11	3.46	2

MW: molecular weight; logP: logarithm of octanol–water partition coefficient; nON: number of hydrogen bond acceptors.

**Table 2 ijms-22-08949-t002:** Fluorescence quantum yield of compounds **1**–**6** in CH_3_OH/H_2_O mixed solution with various water fractions.

Compound	Solvents	Quantum Yields (ϕF)
**1**	CH_3_OH	0.03
	CH_3_OH/H_2_O (3:7)	0.59
	CH_3_OH/H_2_O (1:9)	0.34
**2**	CH_3_OH	0.01
	CH_3_OH/H_2_O (3:7)	0.19
	CH_3_OH/H_2_O (1:9)	0.12
**3**	CH_3_OH	0.07
	CH_3_OH/H_2_O (2:8)	0.21
	CH_3_OH/H_2_O (1:9)	0.15
**4**	CH_3_OH	0.02
	CH_3_OH/H_2_O (4:6)	0.18
	CH_3_OH/H_2_O (1:9)	0.10
**5**	CH_3_OH	0.04
	CH_3_OH/H_2_O (4:6)	0.08
	CH_3_OH/H_2_O (1:9)	0.05
**6**	CH_3_OH	0.04
	CH_3_OH/H_2_O (5:5)	0.14
	CH_3_OH/H_2_O (1:9)	0.09

**Table 3 ijms-22-08949-t003:** Corresponding R values of compounds **1**–**6**.

Compound	Corresponding R Values (Pearson Correlation Coefficient)
**1**	0.87
**2**	0.83
**3**	0.87
**4**	0.88
**5**	0.86
**6**	0.85

## Data Availability

Not applicable.

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
