# Peer review of "Synthesis of New AIEE-Active Chalcones for Imaging of Mitochondria in Living Cells and Zebrafish In Vivo"

_ijms, 2021, doi:10.3390/ijms22168949_

Round 1
Reviewer 1 Report
Review of the article "Synthesis of New AIEE-Active Chalcones for Imaging of Mitochondria in Living Cells and Zebrafish in Vivo" by Feng He et al. Very well planned research, both in terms of spectroscopic research and biological research. An interesting topic, worth extending further, with new chemical compounds. The article is a pleasure to read. I found only small mistakes that did not disturb the scientific tone of the work.
Comments:
1) Were compounds 1-11 new chemicals?
2) Please add Lipiński's rule of five parameters of compounds 1-11 like in article by Kozłowska et al (DOI: 10.3390 / molecules24224129)
3) Figure 1: "(a)" and "(b)" should be smaller.
4) Line 90: should be "CH3OH" not "CH3OH"
5) Figure 2: Graphs in the upper right corners are not clear to reader, they are too small.
6) Line 128: should be "UV-Vis" not "UV-vis".
7) Line 277: should be "methanol" not "Methanol"
8) Line 282: mnissing space between "saline(PBS)"
9) Please add to all compounds their TLC Rf values ​​in proper eluent and UV-Vis band values.
10) Part after section 3.2.11 about spectroscopic characterization of compounds should be done as a separate section - it will make the reading easier for the reader.
11) Line 481: missing space between "serum(FBS)"
Summarizing: the article can be published after fulfilling the remarks mentioned in minor revision.
Reviewer 2 Report
Manuscript ID: ijms-1325199
The article titled “Synthesis of New AIEE-Active Chalcones for Imaging of Mitochondria in Living Cells and Zebrafish in Vivo” by Feng He et al have demonstrated the synthesis of chalcones and studied their aggregation-induced emission enhancement (AIEE) properties. Their excellent fluorescence properties, large stokes shift, and specific target to mitochondria (in-vitro) and digestive tract of zebrafish embryo (in-vivo) were employed for cell imaging. The additional benefits were low cytotoxicity and good biocompatibility at the experimental concentration used.
This manuscript is well written and is scientifically sound. A lot of chemical and biochemical data is generated and therefore this makes it valuable to the scientific community working in the field of fluorescent chalcones for diagnostic application.
General comments-
English language editing is required throughout the manuscript.
In the introduction, line 53-55 authors state that “Therefore we tried to reduce the planarity and molecular weight by breaking the oxygen-containing pentacyclic ring at the end of sanguinarine. The experiments results showed chelerythrine was also typical AIEE molecule.” In fact the addition of one more methyl group in chelerythrine will increase the MW (348 instead of 332 for sanguinarine). This sentence should be revised accordingly.
In the introduction (lines 49-60) the rationale for design of chalcones is directed from the author’s own lab study starting from sanguinarine to chelerythrine and then flavones and flavanone to chalcones (Refs 21, 24 and 25). Several chalcones based AIEE papers already existing in the literature are not cited. Please see individual papers-
https://pubs.rsc.org/en/content/articlelanding/2016/nj/c6nj01387b#!divAbstract
https://www.sciencedirect.com/science/article/abs/pii/S1010603017303350
https://link.springer.com/article/10.1007/s10895-021-02711-6
and revise the introduction accordingly.
The push-pull design strategy discussed/used in these above papers has not been elaborated/used in the designing of 11 chalcones described. Therefore, on that basis a discussion of the structural aspects and AIEE behaviour relationship of chalcones in the current manuscript is required. The presented discussion in lines 136-141 will benefit from the above exercise. Please also include relevant citations for the statement at line 139-41. “It indicates that the fluorescence color of chalcone derivatives can be successfully adjusted and controlled by changing substituent functional groups.”
Chalcones scaffold is vastly explored and relevant citations 26-29 are very limited. Please cite additionally relevant citations like-
https://pubs.acs.org/doi/10.1021/acs.chemrev.7b00020
https://link.springer.com/article/10.1007/s11101-014-9387-8
https://pubmed.ncbi.nlm.nih.gov/28914193/
https://www.sciencedirect.com/science/article/abs/pii/S0223523406003540
Section 3 to be renamed as Materials and methods
Under section 3.2- Synthesis of chalcones, provide one general procedure applicable to all chalcones. The relevant details like recrystallizing solvent can be mentioned along with melting point and NMR data.
Section 3.3.1 needs some revision for example, Compounds 1-11 were weighted accurately by electronic scales: (0.50, 0.50, 0.56, 0.53, 0.56, 0.53, 0.44, 0.53, 0.47, 0.47, 0.50 mg.) Then they were dissolved in methanol solution (x mL) respectively, and mixed well. The final concentration of each solution was 2.07×10-4 M.”
Section 3.6-
“compounds 1-6 with different concentrations (1, 3, 6, 12 μM) were added.” What solvent was used for dissolving these chalcones 1-6? Was the blank solvent tested for cytotoxicity effects at same loading?
Section 3.7- mention which solvent was used to make the 10 µM solution for cell imaging studies.
The concentrations tested in vitro were (1, 3, 6, 12 μM) and for cell imaging 10 µM was used. How was the concentration for in-vivo studies (15 µM) chosen?
